# The Endocannabinoid System in Pediatric Inflammatory and Immune Diseases

**DOI:** 10.3390/ijms20235875

**Published:** 2019-11-23

**Authors:** Maura Argenziano, Chiara Tortora, Giulia Bellini, Alessandra Di Paola, Francesca Punzo, Francesca Rossi

**Affiliations:** 1Department of Experimental Medicine, University of Campania Luigi Vanvitelli, 80138 Naples, Italy; maura.argenziano@unicampania.it (M.A.); giulia.bellini@uniroma3.it (G.B.); alessandra.dipaola@unicampania.it (A.D.P.); francesca.punzo19@gmail.com (F.P.); 2Department of Women, Child, and General and Specialized Surgery, University of Campania Luigi Vanvitelli, 80138 Naples, Italy; chiara.tortora@unicampania.it

**Keywords:** CB1 receptor, CB2 receptor, endocannabinoid system, inflammation, immune regulation

## Abstract

Endocannabinoid system consists of cannabinoid type 1 (CB1) and cannabinoid type 2 (CB2) receptors, their endogenous ligands, and the enzymes responsible for their synthesis and degradation. CB2, to a great extent, and CB1, to a lesser extent, are involved in regulating the immune response. They also regulate the inflammatory processes by inhibiting pro-inflammatory mediator release and immune cell proliferation. This review provides an overview on the role of the endocannabinoid system with a major focus on cannabinoid receptors in the pathogenesis and onset of inflammatory and autoimmune pediatric diseases, such as immune thrombocytopenia, juvenile idiopathic arthritis, inflammatory bowel disease, celiac disease, obesity, neuroinflammatory diseases, and type 1 diabetes mellitus. These disorders have a high social impact and represent a burden for the healthcare system, hence the importance of individuating more innovative and effective treatments. The endocannabinoid system could address this need, representing a possible new diagnostic marker and therapeutic target.

## 1. Introduction

The endocannabinoid (EC) system is a neuromodulator system composed of endogenous cannabinoids, anandamide (AEA) and 2-arachidonoylglycerol (2-AG), their selective receptors cannabinoid receptor type 1 (CB1) and cannabinoid receptor type 2 (CB2), and all the enzymes involved in their synthesis and degradation (N-acyl phosphatidylethanolamine phospholipase D (NAPE-PLD), Monoacylglycerol lipase (MAGL), Diacylglycerol lipase (DAGL), Fatty acid amide hydrolase (FAAH)) [1]. The CB1 receptor is principally expressed in the central nervous system, in particular, on brain presynaptic neurons [2] and spinal cord and dorsal root ganglia [3], while the CB2 receptor is mostly localized on immune peripheral cells, such as B lymphocytes, macrophages, mast cells, natural killer cells, and on the lymphatic organs, spleen, tonsils, and thymus [4]. The EC system is involved in many biological functions: regulation of appetite, pain management, and also organism development since the earliest stage of gestation [5]. Its capability to modulate inflammation and the immune response is surely noteworthy, considering the great clinical relevance that the regulation of these processes could have in several pathologies.

This review aims to give an overview on the role of cannabinoid receptors and other EC elements in pediatric inflammatory and immune diseases, such as immune thrombocytopenia, juvenile idiopathic arthritis, inflammatory bowel disease, celiac disease, obesity, neuroinflammatory diseases, and type 1 diabetes mellitus. All these pathologies share an important characteristic—they are an important burden for the healthcare system and strongly compromise the life quality of patients and their families. For this reason it is very important to identify innovative and effective therapeutic targets, and so far the EC system seems to address this need well.

## 2. The EC System in Inflammation and the Immune Response

The EC system has an important role in maintaining immune system homeostasis as well as in modulating inflammatory processes (Figure 1). Both CB1 and CB2 receptors play this crucial role, but their different localization could explain the different involvement in these biological processes. CB1 receptor exerts its function principally in the nervous system, where it influences the neurotransmitter release at axonic terminals and acts as an anti-inflammatory mediator, restoring, for example, the levels of interleukin 1 beta (IL-1β) and cyclooxygenase-2 (COX2) after inflammatory stimuli, as observed in rats by Zhang et al., 2016 [6].

CB2 has a more consistent role in the peripheral regions, where it principally influences the immune response. A variant of CB2 encoding gene (*rs35761398*) has been individuated that leads to the production of a less functional receptor variant, CB2 Q63R. In the literature, it is well documented that the RR homozygote subjects are more prone to developing autoimmune disorders compared with QQ homozygote subjects [7].

Even though the underlying biological mechanisms need to be better clarified, these functions seem to be associated with the EC system’s capability to inhibit immune cell proliferation and pro-inflammatory mediator release (cytokines, reactive oxygen species (ROS), nitric oxide, etc.) [8]. Cytokines are molecules that positively or negatively mediate inflammation from its initiation to resolution and are produced by macrophages in the earliest stage of innate immune response and by T-cells during the adaptive immune response [9]. Inflammation normally occurs as a defense mechanism and it is self-limiting thanks to an interplay between immune cells and several kind of cytokines [10]. When immune tolerance is compromised, as is observed in autoimmune diseases, this physiological equilibrium is altered in favor of an inflammatory condition, which leads to tissue damage [11]. In the literature, it is reported that the cannabinoids exert a modulatory effect on the release of several cytokines. For example, AEA attenuates the inflammation by reducing the production of the pro-inflammatory IL-6 and nitric oxide from lipopolysaccharide (LPS)-activated macrophages in vitro [12]. In accordance, Klein et al. demonstrated that tetra-hydro-cannabinol (THC), a CB1 and CB2 receptor partial agonist, inhibits the production of IL-12 and interferon gamma (IFN-γ) by type 1 T helper (Th1) cells, thus showing an anti-inflammatory and immunosuppressive effect [13]. Already in 1995, McCoy et al. observed that THC is able to suppress the antigen presentation and, as consequence, the activation of T helper cells [14]. The metabolism of the ligand 2-AG leads to the production of arachidonic acid, which is a precursor of proinflammatory effectors, such as prostaglandins and leukotrienes [15]. In 2014, Sardinha et al. used an inhibitor of MAGL in vivo, the main 2-AG degrading enzyme, observing a reduction in prostaglandins and leukotrienes production with a consequent anti-inflammatory effect [16]. They also observed a reduction in adherent leucocytes number when both MAGL and FAAH were inhibited, thus supporting the anti-inflammatory role of CB2 activation.

In general, the drugs acting on the EC system show an inhibitory effect on the immune system, causing a reduction of B and T lymphocyte proliferation [17]; inhibition of antibody production by B lymphocytes [18]; reduction of chemokine and cytokine production by NK cells [19,20]; attenuation of migration, phagocytosis, and proinflammatory cytokine release in macrophages; and the enhancement of Mesenchymal Stromal Cells’ (MSCs) homing, immunosuppressive and anti-inflammatory activities [21,22,23].

## 3. The EC System in Immune Thrombocytopenia

Immune thrombocytopenia (ITP) is an autoimmune and multifactorial disease in which autoantibodies prematurely disrupt platelets [24]. It can be primary (idiopathic) or secondary to other pathologies. Taking into account the immunomodulating role of the CB2 receptor and the documented correlation between its Q63R variant and autoimmune disorders, in 2011, Rossi et al. genotyped, for the first time, 190 ITP Italian children for the *CNR2* rs35761398 variant [25]. In the following years, Mahmoud et al. and Ezzat et al. performed similar association studies in an Egyptian ITP child population [26,27]. In both study populations, a strong correlation between Q63R polymorphism and the susceptibility to childhood ITP emerged. In addition to this genetic predisposition, an impairment in T-cells together with a direct cytotoxic effect of these cells [28,29,30,31] could have a key role in the still-unclear ITP pathogenesis. The literature reports the presence of activated platelet-autoreactive T cells in ITP patients and also a high Th1/Th2 ratio [32] with a related alteration of cytokine release.

In these patients, Th1 cells produce an excess of IL-2 and INF-γ, which further exacerbates the activity of T-cells [33]. Th2 cells reduce the production of the anti-inflammatory cytokine IL-10, this inhibits the cytotoxic T-cell and B-cell responses [34,35]. The possible involvement of this cytokine in ITP has been deeply investigated over the years. In particular, several authors found a correlation between the IL-10 (-1082) polymorphism, which is associated with an impairment in general cytokine release, and the acute form of ITP [36,37,38,39,40]. It is known that both CB1 and CB2 receptors can influence the production of IL-10. Indeed, it has been reported that the selective stimulation of both receptors promotes an increase in the levels of this cytokine [41,42], supporting the well-described anti-inflammatory properties of the EC system. Among the alterations observed in ITP patients, there is also an impairment in the immune modulatory functions of Mesenchymal Stromal Cells (MSC). They are normally able to inhibit T- and B-cell proliferation [43], whereas in ITP patients, this property is compromised together with their own proliferation [44,45]. It has been observed that the CB2 receptor is more frequently expressed in healthy MSCs than in MSCs obtained from ITP pediatric patients [23] and that a selective stimulation of the cannabinoid receptor can restore the immunomodulatory capabilities of MSCs [46]. Another well-reported aspect in the literature is the capability of AEA and 2-AG to modulate the platelets’ function and survival. Indeed, it has been observed that while AEA inhibits apoptosis in platelets [47,48], 2-AG works as a megakaryopoietic agent [49]. In 2014, Gasperi et al. performed a study on human megakaryocytic MEG-01 cells, observing that 2-AG stimulates MEG-01 maturation and also enhances platelet production. When they inhibited MAGL, the main 2-AG degrading enzyme, these effects are even more evident [50]. All of this evidence provides new insights to understand the ITP-causing mechanisms, its clinical presentation, and also to manage its outcomes.

## 4. The EC System in Juvenile Idiopathic Arthritis

Juvenile idiopathic arthritis (JIA) is a particular form of rheumatoid arthritis (RA), an autoimmune pathology normally affecting adults, that appears in subjects younger than 16 years-old [51]. In general, RA is characterized by synovium inflammation, joint pain, and bone disruption [52]. Therefore, proper treatment for this kind of pathology should avoid inflammation, contain bone destruction, and re-balance the immune response. Targeting the EC system could address these needs, also taking into account the presence in RA synovial fluid of CB receptors together with AEA, 2-AG, and the enzyme FAAH [53]. In particular, several authors have reported that the inhibition of FAAH with selective drugs reduces joint inflammation in many kinds of arthritis, including RA [53,54]. In 2018, Falconer et al. observed that JWH-133, a CB2 selective agonist, causes a switch in macrophage phenotype from the pro-inflammatory M1 to the anti-inflammatory M2, counteracting the inflammation in collagen-induced arthritis (CIA) mice [55]. Also, the knockdown of CB2 in human RA fibroblasts with siRNA inhibits the inflammatory process, causing a reduction of pro-inflammatory cytokines, such as tumor necrosis factor alpha (TNFα), IL-1β, IL-6, and IL-8 [56]. The same group of cytokines is reported to be influenced by 4Q3C, another CB2 selective agonist that reduces their levels, thus showing a strong anti-inflammatory effect and its ability to reduce the osteoclast (OCs) number and activity in CIA mice [57]. This is additive evidence about the well-known CB2 capabilities to counteract both inflammation and bone erosion [58,59,60]. The latter is linked to the reduction of the receptor activator of nuclear factor kappa B ligand (RANKL) usually observed after activation of the CB2 receptor [61].

RANKL is an important osteoclastogenic mediator, physiologically released from osteoblasts (OBs) and able to activate OCs. Any alteration of this pathway could cause a break in the deposition–resorption equilibrium in bone tissue [62]. In RA, an increase in RANKL is in fact observed [63], and it is known that the EC system is present in RA synovial tissue and fluid [53], but not in healthy joints [64]. Also, for this pathology, the anti-inflammatory function of the EC system is crucial and moreover can be associated to the modulation of immune response. Bellini et al. observed that, in a population of 171 JIA children, patients with a RR genotype for the CNR2 gene have an increased risk to develop this pathology [65]. Taken together, all this evidence suggests a crucial role for CB2 receptor in pathogenesis and clinical course of both RA and, in particular, JIA.

## 5. The EC System in Inflammatory Bowel Disease and Celiac Disease

Inflammatory bowel diseases (IBD) are immune-mediated inflammation conditions affecting the gastro-intestinal tract, including Crohn’s disease (CRD) and ulcerative colitis (UC) [66]. Several studies suggest that gut inflammation is associated with hyper-expression of cannabinoid receptors [67], to increased levels of the AEA [68], that suppresses T-cell proliferation and inhibits IL-2, TNF-α, and INF-γ release from activated T-lymphocytes [69] and also with alteration in enzymatic pathways. In particular, in 2004, Massa et al. observed that mice deficient in FAAH, the principal AEA-degrading enzyme, are protected from colitis, and the pharmacological blockade of the CB1 receptor leads to a worsening of murine colitis [70]. Also, 2-AG and the enzyme responsible for its degradation, MAGL, are involved in gut inflammation. Indeed, it has been observed that after treatment with a selective MAGL inhibitor, there is a reduction of colon inflammation in mice with induced colitis [71]. Wright et al. demonstrated increased CB2 expression in the epithelium of colonic tissue in the acute phase of IBD and a reduced secretion of IL-8, suggesting an anti-inflammatory effect of the receptor [72]. In accordance with this Ihenetu et al. demonstrated the same effect of CB2 activation in colonic epithelial cell lines [73]. Recently, Leinwand et al. demonstrated that CB2 stimulation ameliorates inflammation in a mouse model of CRD [74].

Moreover, it is widely known that the EC modulation ameliorates the IBD symptomatology reducing nausea, abdominal pain, and diarrhea [75]. Therefore, CB2 has a double potential—to reduce the inflammation and to improve the IBD patients’ life quality. IBD is also characterized by an immune component, but in the literature, the opinions about an eventual genetic predisposition related to the Q63R variant of CB2 are discordant. While in 2014, Yonal et al. demonstrated the absence of association between this variant and IBD in a Turkish population [76], a few years later, in 2018, Strisciuglio et al. demonstrated that the R63 variant is associated with a severe phenotype in both UC and CRD Italian pediatric patients [77]. Race and ethnicity strongly influence the IBD incidence and phenotype [78], potentially explaining the different distribution of the CB2 Q63R variant between Turkish and Italian populations.

Celiac disease (CD) is also described as an inflammatory and autoimmune disease of the small bowel that occurs in genetically predisposed subjects after the ingestion of gluten [79]. Rossi et al. demonstrated, in a cohort of 327 CD children, a significant association between the CB2 Q63R variant and CD, confirming the role of the receptor in autoimmunity susceptibility [80]. There are several in vivo and in vitro studies demonstrating an alteration of the EC system in untreated CD patients [80,81,82]. In detail, in the active mucosa of these patients, an increase in AEA levels has been observed [68], which is very likely due to the high expression of NAPE-PDL, the main enzyme responsible for AEA synthesis, observed in vivo by Battista et al., 2012 [81]. In untreated celiacs, celiacs on a gluten-free diet, and controls, the same authors did not observe any variation in FAAH, which is instead the principal AEA-degrading enzyme. Moreover, it an increase in the expression of the CB2 receptor has been observed both in the duodenal mucosa and in the atrophic villous during active disease, whereas CB receptor levels are normal in treated CD patients. It is probable that the increase in CB2 levels is due to lymphocyte infiltration, in particular, to CD4+ T-cells [83] that produce the pro-inflammatory component IL-17. Therefore, the selective stimulation of CB2, by means of AEA for example, could exert an immunosuppressive effect reducing IL-17 production. Interestingly, the ex vivo incubation of celiac biopsies with gliadin increases the expression of both receptors [80,84]. In conclusion, a deregulation of the EC system could be implicated in the pathogenesis of CD but further studies are certainly needed to clarify its therapeutic role.

## 6. The EC System in Obesity and Fatty Liver Disease

Obesity is one of the most diffused diseases worldwide and represents an important public health problem. In recent years, a critical increase of childhood obesity has been observed, the most common cause of which is excessive caloric intake compared with the actual caloric expenditure in genetically predisposed subjects [85]. It is characterized by a low-grade inflammation in white adipose tissue, which leads to the production and secretion of inflammatory mediators responsible for attracting macrophages and other cells of the immune system [86]. In particular, over the years, many authors investigated the possible connection between the pro-inflammatory mediators TNF-α and IL-6 and obesity in children. Data present in the literature are discordant. Indeed, while in several study populations, there are no differences in these cytokine levels between normal- and over-weight children [87,88], in 2008, Cabellero et al. reported raised levels of TNF-α in obese Hispanic children [89]. It has been demonstrated that the EC system modulation is involved in regulating obesity [90,91], but CB1 and CB2 receptors play different roles. While CB1 activation induces increasing inflammatory processes [92], CB2 activation ameliorates the inflammatory state related to obesity [93]. Actually, there are different opinions about the role of CB2 in obesity. Indeed, in 2010, Pacher and Mechoulam described it as a “cannabinoid receptor with an identity crisis” [94]. Some authors observed that 2-month-old CB2 –/– mice under a high-fat diet did not show weight gain, indicating that a lack of CB2 has protective effects [95] and even its stimulation can correlate with an increase in inflammation [96]. However, data suggesting anti-inflammatory properties of CB2 remain more representative. It is well documented, for example, that the CB2 receptor agonist JWH-133 in mice is able to inhibit auto-reactive T cells, thus preventing leukocyte migration into the inflamed tissue [97].

The stimulation of CB2 leads also to another important protective effect in adipocytes from obese subjects: the increase of uncoupling protein 1 (UCP1) levels and consequently of heat generation and energy expenditure [98]. In support of the CB2 protective role, for obesity, a strong association with CB2 Q63R variant has also been observed. Indeed, children expressing this variant show an increase in weight and high levels of pro-inflammatory cytokines (IL-6 and TNF-α). When the CB2 receptor is selectively stimulated with JWH-133, this critical condition is restored in primary cultures of adipocytes obtained from obese subjects [93,98,99]. Another polymorphism positively associated with obesity is the FAAH polymorphism rs324420 that reduces FAAH activity, leading to higher AEA levels and consequent over-activation of CB1 receptors, which, in turn, causes adipogenesis and a high obesity risk [100,101]. The secretion of pro-inflammatory mediators by adipose tissue can increase the risk of developing secondary pathologies, such as type 2 diabetes, cardiovascular diseases, cancer, and non-alcoholic fatty liver disease (NAFLD) [102]. It has been observed that the CB2 Q63R variant is also present in subjects with NAFLD and is associated with high-grade inflammation in the liver [103]. Since CB1 and CB2 receptors are also expressed in this organ, their regulation could be important for reducing liver impairments, as demonstrated also by Coppola et al. in patients with chronic hepatitis C (HCV) [103,104,105,106,107]. In collecting all of this evidence, it is possible to suggest the EC system as pathological marker and pharmacological target to manage, not only the obesity-associated inflammation, but also its secondary complications.

## 7. The EC System in Neuroinflammatory Diseases

Neuroinflammation is an inflammatory process that aims to defend the brain in case of damage, but it can evolve into a chronic pathological condition. It can be associated with several pathologies, such as Alzheimer’s disease, Parkinson’s disease, schizophrenia, bipolarism, and also, regarding pediatric patients, neurodevelopment disorders, such as autism spectrum disorders and epilepsy [108,109,110]. The CB1 receptor has a widespread distribution in the brain, particularly in the pre-synaptic region on axon terminals [111,112,113] and it seems to be responsible for the protection of neurons from death associated with neuroinflammation [114]. While its stimulation with selective agonists inhibits the release of pro-inflammatory mediators (nitric oxide, TNF-α, and COX-2) [115,116], its ablation up-regulates the microglia activity as observed in CNR1 –/– mice, in which the hippocampus expression of the pro-inflammatory cytokine IL-6 was high [117]. Microglia cells are brain-resident macrophages involved in these pathological conditions and they constitutively express the CB1 receptor (Figure 2). They can be activated as pro-inflammatory M1 and then release cytokines such as TNF-α, IL-6, and IL-1β, or as anti-inflammatory M2 and release inflammation-inhibiting factors such as IL-10. In the literature, the involvement of the FAAH enzyme in modulating microglia cell activation has been widely hypothesized [118,119]. One of the most recent studies about the role of FAAH highlighted that its knockdown in BV2 microglia cells and the related increase of AEA levels leads to an overexpression of M2 markers, revealing an important anti-inflammatory effect [120]. During inflammatory processes, the microglia cells over express the CB1 receptor and release a consistent amount of cytokines and mediators that damage cells.

Several authors have reported that the selective stimulation of the receptor reduces microglia activity, thus exerting neuroprotection [121]. For all these reasons, there is a growing interest in targeting the EC system for treatment of different neurological diseases. Epidiolex is the first cannabidiol-based drug approved by the Food and Drug Administration (FDA) for the treatment of childhood epilepsies, such as Dravet syndrome and Lennox–Gastaut syndrome [122]. As indicated by Devinsky et al., 2015, 137 epileptic children patients who received this drug self-reported a reduction of seizure frequency by 54% without significant side effects (somnolence, decreased appetite, and diarrhea) [123]. Even if it is less present in neuronal compartments, the CB2 receptor could also have a role in neuroinflammation. Siniscalco et al. performed a study using a population of 22 autistic children from which a CB2 up-regulation in peripheral blood mononuclear cells (PBMCs) emerged. This observation obviously generates the hypothesis of a possible involvement of the EC system in pathogenesis of autism spectrum disorders and highlights the possibility of using the CB2 receptor as a therapeutic target [124], thus avoiding the psychotropic effects that often derive from CB1 stimulation.

## 8. The EC System in Type 1 Diabetes Mellitus

Type 1 diabetes mellitus (T1DM) is an autoimmune metabolic disorder due to the destruction of insulin-producing pancreatic β-cells by a specific auto-antibody that, over the years, strongly compromises patients’ life quality [125]. Despite T1DM being defined as a chronic autoimmune disease, there is evidence about a crucial role of inflammation and oxidative stress in its pathophysiology. In 2016, Domingueti et al. observed that inflammatory biomarkers, such IL-6 and TNF-α, are increased in diabetic patients with micro- and macro-vascular complications [126]. Both human and rat pancreatic β-cells express the CB2 receptor [127]. Considering the CB2 capacity to reduce pro-inflammatory mediators when properly stimulated [128], there is a growing interest in using it as pharmacological target to contain the effects of the disease [129,130].

In the literature, there are several preclinical studies performed on rodents with streptozotocin (STZ, a pancreatic β-cell cytotoxin) induced T1DM, in which it has been demonstrated how the stimulation of CB2 with drugs can ameliorate the symptoms of the disease, such as hyperalgesia [131,132]. Moreover Zoja et al. observed that in kidney biopsies from patients with diabetic nephropathy, CB2 is down-regulated, and moreover, its deletion worsens the kidney state, confirming the receptor protective action [129]. Moriarty et al. observed that STZ-diabetic rats show an alteration in CB1 receptor functionality in the substantia nigra, which may be associated with diabetes and diabetic neuropathic pain [133]. Surely all of this evidence needs further investigation, but new perspectives for the management of this complex degenerative disease are surely emerging.

## 9. Conclusions

Starting from the well-known EC system capabilities to modulate inflammation and the immune response, many authors have demonstrated its involvement in the pathogenesis and in the onset of several childhood pathologies (immune thrombocytopenia, juvenile idiopathic arthritis, inflammatory bowel disease, celiac disease, obesity, neuroinflammatory diseases, and type 1 diabetes mellitus) (Table 1). They are able to influence the production of inflammatory mediators by mechanisms that are still unclear. In particular, their proper stimulation inhibits the secretion of pro-inflammatory factors and also reduces the activation and the toxic effects of the immune cells. It clearly emerges from this review that CB2 receptors could be the best pharmacological target. Indeed, because of its peripheral localization, its stimulation is associated with a lower risk of psychotropic side effects compared with the stimulation of CB1. This aspect is crucial in supporting the possibility to introduce the CB2 selective agonists in the therapeutic protocols as safe and effective additive drugs. This innovative intervention could represent a strong aid to ameliorate not only the onset of these pathologies but also the life quality of affected children.

Moreover, it is important to remember that the diseases explored in this review are characterized not only by their primary symptomatology but also by secondary consequences, which will certainly have an impact on the health of ‘future’ adults. Furthermore, avoiding a lifelong follow-up of these patients, at least for comorbidities, should be a great clinical success and also reduce the related costs for the healthcare system.

The Table 1 summarizes the main alterations of the endocannabinoid receptors (CB1 and CB2), their endogenous ligands (AEA and 2-AG), and the enzymes responsible for their synthesis and degradation (NAPE-PLD, FAAH, MAGL) in pediatric inflammatory and immune diseases.

## Figures and Tables

**Figure 1 ijms-20-05875-f001:**
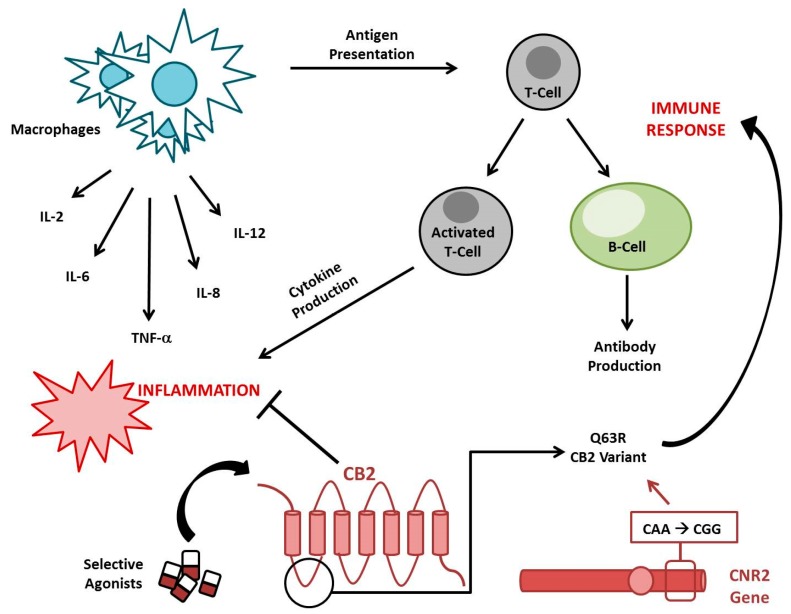
Role of Cannabinoid Receptor 2 (CB2) in inflammation and the immune response. The stimulation of CB2 by its selective agonists inhibits cytokines’ production and reduces antigen presentation, modulating both inflammation and the immune response. CNR2 Q63R is a very common CB2 variant that compromises CB2 immunomodulatory properties, predisposing the individual to autoimmune disorders.

**Figure 2 ijms-20-05875-f002:**
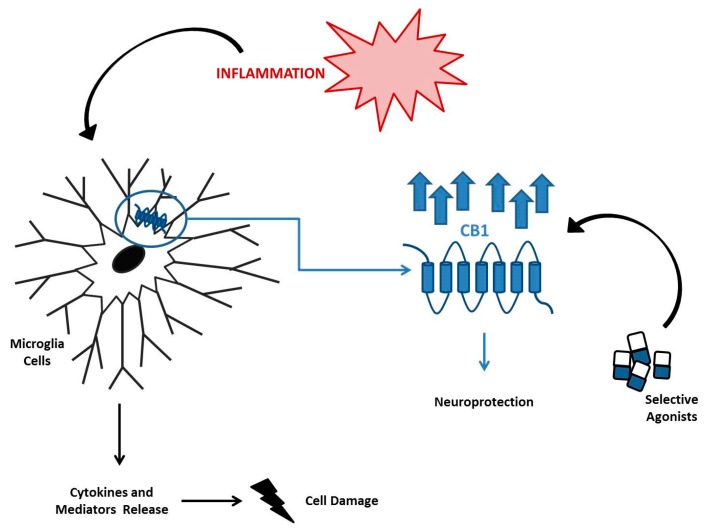
Role of Cannabinoid Receptor 1 (CB1) in Neuroinflammatory Diseases. During the inflammatory process, microglia cells overexpress CB1 and are responsible for the release of cytokines and toxic mediators. CB1 stimulation by its selective agonists has a protective role in neurons, counteracting neuroinflammation and its associated diseases.

**Table 1 ijms-20-05875-t001:** EC system main alterations in pediatric inflammatory and immune diseases.

Disease	Main Alterations in the EC System
Immune thrombocytopenia	Increased risk by *CNR2* rs35761398 variant; reduced CB2 receptor expression in Mesenchymal Stromal Cells (MSC)
Juvenile idiopathic arthritis	Increased risk by *CNR2* rs35761398 variant; presence of EC elements in synovial fluid; inflammation; RANK-L accumulation in joints
Inflammatory bowel diseases	Increased risk by *CNR2* rs35761398 variant; CB receptors hyper-expression; increased AEA levels
Celiac Disease	Increased risk by *CNR2* rs35761398 variant; increased NAPE-PDL levels; hyper-expression of CB2 receptor in duodenal mucosa and in the atrophic villous
Obesity and NAFLD	Increased risk by FAAH rs324420 polymorphism and *CNR2* rs35761398 variant
Neuroinflammatory Diseases	Altered expression of CB receptors
Type 1 diabetes mellitus	Altered functionality of CB1 receptor in substantia nigra; CB2 receptor down-regulation

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
