# Peer review of "The Endocannabinoid System in Pediatric Inflammatory and Immune Diseases"

_ijms, 2019, doi:10.3390/ijms20235875_

Round 1
Reviewer 1 Report
The review "Endocannabinoid system in pediatric inflammatory and immune diseases" by Dr. Argenziano and coworkers describes the role of cannabinoid receptors in some pathologies with particular focus on those diagnosed in childhood. The originality is good and the overall presentation is fair.
Despite the appealing title, the data reported in the text does not fit perfectly with this. In fact, among the several elements that form the endocannabinoid system, the information are focused only on the involvement of cannabinoid receptors in the inflammatory diseases, excluding the alterations of all enzymes responsible for the metabolic routes. I strongly recommend to improve the paper by adding the lacking data (if available in literature).
I also suggest to add a table in order to summarize the data reported in the sections.
The english language needs minor corrections probably due to typing error.
Author Response
Response to Reviewer 1 Comments.
We thank the reviewer for his/her comments and for the opportunity to improve our manuscript.
Point 1. “Despite the appealing title, the data reported in the text does not fit perfectly with this. In fact, among the several elements that form the endocannabinoid system, the information are focused only on the involvement of cannabinoid receptors in the inflammatory diseases, excluding the alterations of all enzymes responsible for the metabolic routes. I strongly recommend to improve the paper by adding the lacking data (if available in literature)”
Response 1. In literature, the involvement of the endocannabinoid system in inflammation and immune response is well described with particular attention on the receptors’ role. As suggested, we reviewed once again the literature in search for papers about the involvement of all enzymes responsible for the metabolic routes in inflammatory diseases, but no relevant data worth to be mentioned are present.
Point 2. “I also suggest to add a table in order to summarize the data reported in the sections”
Response 2. Our manuscript is a review of the last years’ literature about the involvement of endocannabinoid receptors in inflammatory and immune processes and no experimental data are reported, therefore we would find difficult to fit them in a table.
Point 3. “The english language needs minor corrections probably due to typing error”
Response 3. We corrected the English errors.
Reviewer 2 Report
The authors have summarised the role of the ECS in immune diseases.
The authors should be quite clear from the onset, that the main receptor involved would be CB2 due to its localisation, and the role of CB1 is much less.
More specific comments:
Abstract
Line 11: composed is not the appropriate term
I think in the abstract you need to be clear, CB2 to a great extent and CB1 to a lesser extent are involved in regulating the immune response
This review provides an overview on the role of Endocannabinoid system in pathogenesis and onset of inflammatory and autoimmune pediatric diseases, such as Immune Thrombocytopenia, Juvenile Idiopathic Arthritis, Inflammatory Bowel Disease, Celiac Disease, Obesity, Neuroinflammatory Diseases and Type 1 Diabetes Mellitus.- a number if these terms should not start with a capital letter. This also occurs throughout the manuscript. Diseases do not start with a capital letter mid-sentence
Introduction
individuate innovative - this is not clear what this means
2. EC system...
At the beginning of this section there needs to be a clear statement that CB2 is the main driver
remove the term i.e. throughout and use proper sentence structure
60-what receptors or receptor does THC bind to?
CB1 receptor exerts its function principally in nervous system where it acts as anti-inflammatory mediator,-how does CB1 act as an anti-inflammatory mediator?
Figures are clear and needed
3.
disrupt platelets. -disrupt platelets how?
Q63R 92 polymorphism -as this is a change in protein sequence and linked to disease isnt it a mutation?
that CB1 blockade with a selective antagonist can cause a reduction in the levels of this cytokine-does this mean it is pro-inflammatory?
Besides all these experimental evidences-please rephrase
Also the Celiac Disease-remove the; this sentence also needs a reference
6
there is a lot of llterature which was not discussed here
CB1 receptor has a widespread distribution in brain, particularly in the pre-synaptic region on axon terminals [97-99] and it seems to be responsible for protection of neurons from death associated to neuroinflammation-this does not make it anti-inflammatory; there needs to be better establishment of the role of CB1 in inflammation
Author Response
Response to Reviewer 2 Comments.
We thank the reviewer for his/her comments and for the opportunity to improve our manuscript.
Point 1. “The authors have summarised the role of the ECS in immune diseases. The authors should be quite clear from the onset, that the main receptor involved would be CB2 due to its localisation, and the role of CB1 is much less”
Response 1. In this revised version of our manuscript we explained more clearly from the beginning that CB2 receptor is more involved in immune response modulation compared to CB1.
Point 2. “Line 11: composed is not the appropriate term”
Response 2. We changed it in “consists of”.
Point 3. “I think in the abstract you need to be clear, CB2 to a great extent and CB1 to a lesser extent are involved in regulating the immune response”
Response 3. We underlined the difference in the role of CB2 and CB1 in modulating the immune response as you suggested (Lines 12 and 13).
Point 4. “This review provides an overview on the role of Endocannabinoid system in pathogenesis and onset of inflammatory and autoimmune pediatric diseases, such as Immune Thrombocytopenia, Juvenile Idiopathic Arthritis, Inflammatory Bowel Disease, Celiac Disease, Obesity, Neuroinflammatory Diseases and Type 1 Diabetes Mellitus.- a number if these terms should not start with a capital letter. This also occurs throughout the manuscript. Diseases do not start with a capital letter mid-sentence”
Response 4. We corrected the above mentioned words, as you suggested.
Point 5. “individuate innovative - this is not clear what this means”
Response 5. We changed the term “individuate” with “identify” because we meant to say that the identification of innovative therapies is necessary to ameliorate the life quality of patients and to lighten the burden on the Healthcare System.
Point 6. “2. EC system… At the beginning of this section there needs to be a clear statement that CB2 is the main driver”
Response 6. We moved a part of this section (from line 70-74 to line 55-59) in order to better clarify the role of CB2 receptor.
Point 7. “remove the term i.e. throughout and use proper sentence structure”
Response 7. We correct the structure of the sentences avoiding i.e., as suggested.
Point 8. “60-what receptors or receptor does THC bind to?”
Response 8. We added that THC is a partial agonist of both CB1 and CB2 receptors (lines 72 and 73).
Point 9. “CB1 receptor exerts its function principally in nervous system where it acts as anti-inflammatory mediator-how does CB1 act as an anti-inflammatory mediator?”
Response 9. In the revised version of the manuscript, from line 233 to line 236, we clarified how CB1 receptor exerts its anti-inflammatory activity.
Point 10. “Figures are clear and needed”
Response 10. Thanks for the appreciation.
Point 11. “disrupt platelets. -disrupt platelets how?”
Response 11. Immune thrombocytopenia (ITP) is a characterized by increased platelet clearance, via anti-platelet autoantibodies that in particular target platelet surface proteins. Integrin αIIbβ3 and glycoprotein Ib-IX (GPIb-IX) are the most highly expressed receptor complexes on the platelet surface and represent the main targets of these autoantibodies. (“Fc-independent immune thrombocytopenia via mechanomolecular signaling in platelets.” Quach ME et al. Blood. 2018 Feb 15;131(7):787-796. doi: 10.1182/blood-2017-05-784975. Epub 2017 Dec 4.)
Point 12. “Q63R 92 polymorphism -as this is a change in protein sequence and linked to disease isnt it a mutation?”
Response 12. Q63R variant is commonly defined as polymorphism or SNP because it does not directly lead to a disease, but it is only linked to a predisposition to develop autoimmune diseases. Moreover, it has an allele frequency above 1% in different populations.
Point 13. “that CB1 blockade with a selective antagonist can cause a reduction in the levels of this cytokine-does this mean it is pro-inflammatory?”
Response 13. We made indeed a mistake. In the revised version of the manuscript we corrected the sentence (Lines 115-118).
Point 14. “Besides all these experimental evidences-please rephrase”
Response 14. As suggested, we rephrased this sentence.
Point 15. “Also the Celiac Disease-remove the; this sentence also needs a reference”
Response 15. We removed the article and added a reference.
Point 16. “there is a lot of literature which was not discussed here. CB1 receptor has a widespread distribution in brain, particularly in the pre-synaptic region on axon terminals [97-99] and it seems to be responsible for protection of neurons from death associated to neuroinflammation-this does not make it anti-inflammatory; there needs to be better establishment of the role of CB1 in inflammation”
Response 16. We better clarified the role of CB1 in inflammation, adding dissertation about three papers in which it is well described how CB1 receptor exerts its anti-inflammatory activity (233-236 lines).
Round 2
Reviewer 1 Report
Based on the literature, several relevant and recent papers highlight the involvement of the enzymes that regulate (at least) the AEA metabolism in the pathologies listed in the manuscript. The authors are strongly encouraged to add these data and, again, a Table summarizing the main alterations of the endocannabinoid system in the immune/inflammatory disease.
Author Response
Response to Reviewer 1 Comments.
We thank the reviewer for his/her comments.
Point 1. “Based on the literature, several relevant and recent papers highlight the involvement of the enzymes that regulate (at least) the AEA metabolism in the pathologies listed in the manuscript. The authors are strongly encouraged to add these data […]”
Response 1. As suggested, we improve our manuscript adding recent papers about the involvement of enzymes that regulate AEA metabolism and the others, such as FAAH, MAGL and NAPE-PDL, in the described pathologies (Lines 75-80; 130-136; 144-147; 172-180; 201-205; 242-245; 270-275).
Point 2. “[…] and, again, a Table summarizing the main alterations of the endocannabinoid system in the immune/inflammatory disease”
Response 2. Upon your suggestion, we added a table where we list the main alterations of the endocannabinoid system in the described diseases (Lines 336-341).
Round 3
Reviewer 1 Report
Although the authors could have cited more recent literature data, they revised their paper following the Reviewer' suggestions.
Minor revisions:
P.1, line 17: Please, add "with a major focus on cannabinoid receptors" after "system".
P.1, line 39: Please, modify " on the role of the EC system" with "on the role of cannabinoid receptors and other EC elements".
P.9, line 340:Please, quote first NAPE-PLD (responsible for AEA synthesis) and then FAAH and MAGL (responsible for AEA and 2-AG degradation)
Author Response
Response to Reviewer 1 Comments.
We thank the reviewer for his/her comments.
Point 1. P.1, line 17: Please, add "with a major focus on cannabinoid receptors" after "system"
Response 1. As suggested, we add the indicated sentence (Line 17).
Point 2. P.1, line 39: Please, modify " on the role of the EC system" with "on the role of cannabinoid receptors and other EC elements".
Response 2. We modified the sentence, as suggested (Lines 40-41).
Point 3. P.9, line 340:Please, quote first NAPE-PLD (responsible for AEA synthesis) and then FAAH and MAGL (responsible for AEA and 2-AG degradation)
Response 3. As indicated, we corrected the order of the enzymes (Line 343).